# Neural mechanisms of economic commitment in the human medial prefrontal cortex

**Konstantinos Tsetsos[1]\*, Valentin Wyart[2], S Paul Shorkey[1], Christopher Summerfield[1]**

[1]Department of Experimental Psychology, University of Oxford, Oxford, United Kingdom; [2]Département d'Etudes Cognitives, Ecole Normale Supérieure, Paris, France

**Abstract** Neurobiologists have studied decisions by offering successive, independent choices between goods or gambles. However, choices often have lasting consequences, as when investing in a house or choosing a partner. Here, humans decided whether to commit (by acceptance or rejection) to prospects that provided sustained financial return. BOLD signals in the rostral medial prefrontal cortex (rmPFC) encoded stimulus value only when acceptance or rejection was deferred into the future, suggesting a role in integrating value signals over time. By contrast, the dorsal anterior cingulate cortex (dACC) encoded stimulus value only when participants rejected (or deferred accepting) a prospect. dACC BOLD signals reflected two decision biases–to defer commitments to later, and to weight potential losses more heavily than gains–that (paradoxically) maximised reward in this task. These findings offer fresh insights into the pressures that shape economic decisions, and the computation of value in the medial prefrontal cortex.

\*For correspondence: konstantinos.tsetsos@psy.ox.ac.uk

**Competing interests:** The authors declare that no competing interests exist.

**Reviewing editor**: Eve Marder, Brandeis University, United States

## Introduction

Animals make choices that enhance their chances of positive reinforcement (*Thorndike, 1898*). Laboratory-based tasks have investigated reward-guided decision-making by requiring successive, independent choices to be made in pursuit of a primary reinforcer (e.g., juice) or a flexible resource (e.g., money). For example, on each trial participants might be asked to choose between one of two abstract symbols to obtain a variable monetary reward (*Daw et al., 2006*), or decide which of two snacks they would like to eat upon completion of the experiment (*Lim et al., 2011*). In these tasks, decisions are often characterised by stereotyped biases that hinder outcome maximisation, including a tendency to weight losses more heavily than gains (loss aversion) (*Tversky and Kahneman, 1991*; *Tom et al., 2007*), or an undue preference for an already endowed or 'default' option (status quo bias) (*Kahneman et al., 1991*; *De Martino et al., 2009*; *Fleming et al., 2010*). In conjunction with single-cell recordings (*Tremblay and Schultz, 1999*; *Shidara and Richmond, 2002*; *Padoa-Schioppa and Assad, 2006*; *Hayden et al., 2011*; *Kennerley et al., 2011*) or functional neuroimaging (*Plassmann et al., 2007*; *Basten et al., 2010*; *Hare et al., 2011*; *Lim et al., 2011*; *Hunt et al., 2012*; *Kolling et al., 2012*; *Boorman et al., 2013*), studies have revealed that two interconnected medial cortical regions, the dorsal anterior cingulate cortex (dACC) and the rostromedial prefrontal cortex (rmPFC), play a pivotal role in reward-guided decision-making, although the relative contribution of these regions remains a focus of lively debate (*Kable and Glimcher, 2009*; *Rangel and Hare, 2010*; *Rushworth et al., 2011*).

Tasks involving successive, independent decisions (e.g., standard 'bandit' tasks) allow researchers to simulate key behaviours such as foraging, where an animal makes repeated choices about which

**eLife digest** Humans, in general, are not particularly good at making economic decisions. People can be influenced by unhelpful biases: such as 'loss aversion' where a person views losses as more significant than gains. Sometimes these biases stop us making the decisions that offer the best reward, as such, they raise the question: why do these biases exist at all?

One way to examine this question is by looking at the brain activity of people making economic decisions. Two regions near the front of the brain are known to be involved in human decision-making in response to rewards. However, many researchers disagree as to what these two regions are actually doing when we make economic decisions.

Much of the research in this area has asked participants to essentially gamble on a series of independent events, which typically provide a one-off instant reward with no further positive consequences. However, these tasks do not accurately reflect real economic decisions. In real life situations, people tend to take time to make a decision, and weigh up the potential long-term costs and benefits of an investment. Indeed the decision itself may be deferred until enough information is gathered; for example, very few people would choose to buy a house on the spur of the moment.

Now Tsetsos et al. have attempted to bridge the gap between previous studies and everyday experiences by designing a task that encompasses many of the factors involved in real life decision-making. In this task, participants were given the option of deciding whether to commit to, or reject, an investment opportunity immediately; or to choose to defer making the decision until later—similar to how a person might wait to view different properties before deciding which house to buy. Using brain imaging, Tsetsos et al. found that one of the two brain regions (called the dorsal ACC for short) was involved in weighing up the cost of rejecting an offer, but not accepting it. The other region (called the rostromedial prefrontal cortex or rmPFC) was involved in assessing the value of an offer only when the participant decided to defer making a decision, and not when they decided to commit.

Furthermore, by using computer simulations, Tsetsos et al. found that, with this more realistic task, biases such as loss aversion were in fact beneficial and helped participants to make decisions that increased their financial payoff. This suggests that the 'unhelpful biases' often seen in traditional decision making tasks may be a result of participants' real life strategies failing to work when applied to an artificial situation. In other words, perhaps humans are not so bad at economic decision-making after all.

food item to consume (*Rushworth et al., 2011*). These decisions depend on the momentary utility of a stimulus, that is, the reward that would accrue if that stimulus were to be consumed or disbursed all at once, whether immediately or (as in inter-temporal choice) after a delay (*Kable and Glimcher, 2007*). However, many (perhaps most) economic behaviours are not well captured by this paradigm, because rather than involving successive, independent choices, they require *investment*—that is, long-term commitment to a prospect in anticipation of sustained economic return, and with penalties incurred by any future change of mind. For example, the benefits of choosing the right employment could persist for many years into the future, whereas a poor decision about which mobile telephone to purchase might cause frustration for several months. Other decisions reverse a previous commitment, for example when deciding to sell stock options or to end a failing relationship. In these types of decision, which we refer to as economic 'commitments', prospects are irreversibly 'ruled in' (i.e., by acceptance) or 'ruled out' (i.e., by rejection) of a portfolio of assets that yield sustained positive or negative return to the individual. Unlike the choices made in most current lab-based approaches, economic commitments are not independent: a decision made at a time $t$ continues to contribute to economic return at a later time $t + 1$, and may influence other choices made at that time. The aim of the current work was to understand the computational mechanisms by which economic commitments are made in humans, and to investigate their neural implementation in the reward circuitry of the medial prefrontal cortex.

Commitments often follow a period of deliberation, during which items are considered but final acceptance or rejection is deferred to a later moment (*Shafir and Tversky, 1992*; *Shafir, 1993*). For example, a university student might decide to opt for a course after attending an interesting first

seminar (acceptance, or 'ruling in'); or she might decide to wait until after a second seminar to make a commitment. Equally, the student might decide to drop a course after attending a particularly boring lecture (rejection, or 'ruling out'); or she might give the lecturer another chance, and defer the decision until later. In other words, many behaviours involve choosing between either acceptance and deferral, or rejection and deferral. The notion that deliberation incurs a dual demand associated with selection (what to decide, either option A or B) and commitment (when to decide, either now or later) has received detailed consideration in psychophysical studies, in particular via the modelling of reaction time distributions (*Bogacz et al., 2006*). However, studies requiring fast category judgments make it hard to disentangle the mechanisms determining what to decide and when to commit. In description-based judgment tasks, framing a choice as 'accept' or 'reject' provokes well-described biases in choice behavior (*Tversky and Kahneman, 1981*). However, the issue of how commitments to prospects are made by acceptance or rejection has received less attention in the domain of neuroeconomics (*Furl and Averbeck, 2011*; *Gluth et al., 2012*).

During choices among two or more options with uncertain value, activity in anterior rmPFC has been shown to signal the relative advantage of the chosen or attended option over its competitors (*Padoa-Schioppa and Assad, 2006*; *Lim et al., 2011*; *Hunt et al., 2012*), whereas the dACC often shows the reverse pattern. This may be because dACC preferentially responds to decision entropy or conflict (*Botvinick et al., 1999*), or alternatively because it encodes the value of disengaging from a current or default state to explore a novel course of action (*Hayden et al., 2011*; *Kolling et al., 2012*). In either case however, it remains unknown whether this value difference coding depends on whether stimuli are accepted or rejected, because when deciding between two prospects, an option may be chosen either because it was preferred, or because the alternative was dispreferred. Moreover, it remains unknown how value encoding in the medial prefrontal cortex depends on whether decisions involve economic commitment or not.

Here, thus, we investigated the neural mechanisms that accompany commitment (acceptance, rejection), and deferral (failure to accept or reject) during economic choice, using a multi-alternative choice task in which decisions had financial ramifications that persisted over prolonged episodes, and could not be reversed. In half of the blocks, participants had to choose between accepting (*inclusion by commitment*) and deferring acceptance of a prospect (*exclusion by deferral*). In the other half of the blocks, participants chose between rejecting (*exclusion by commitment*) and deferring rejection (*inclusion by deferral*). Therefore, preference for a bandit would be implied by commitment in rule-in and deferral in rule-out. Our task, thus, allowed us to probe value encoding in the prefrontal cortex as a function of whether a stimulus was preferred (i.e., included or excluded) and whether commitment was made now or deferred until later. Further, the task captured many aspects of economic decisions in the real world: uncertainty about the true value of a prospect, sustained yield accruing from the investment, economic benefit determined collectively by current assets, and the need to trade-off exploration and exploitation.

To preview our findings, whole-brain functional neuroimaging during performance of the task revealed a striking dissociation in the medial prefrontal cortex. The dorsal anterior cingulate cortex (dACC) encoded value when a prospect was excluded (not included) while the rmPFC encoded value only during deferral (not commitment). Furthermore, joint consideration of the behavioural data and the dACC activity allowed us to pinpoint two pressures that shaped decisions in the task: a bias to defer until a later date, and a bias to weight unfavourable (excluded) options more heavily. Although similar biases typically hinder reward harvesting in standard tasks, for our ecologically valid setting we show that they actually allow participants to perform closer to a reward-maximizing agent.

## Results

### Task summary

On each block, participants ($n = 20$, performing the task in the fMRI scanner) viewed spirals of variable length associated with four options (bandits, indexed $i$). On each trial, each spiral of length $s_i$ was drawn from a Gaussian distribution with mean $v_i$ that remained unchanged for that bandit during a block of 12 trials (see *Figure 1A* and 'Materials and methods'). On each trial of a given block, a virtual *pool of assets* contained the preferred bandits thus far. The contents of the asset pool were converted to monetary reward, as we describe below. Each bandit yielded a monetary *payoff* ($\mu_i$) determined by the rank of its mean $v_i$ relative to the mean of all four bandits in the block (longer spirals were

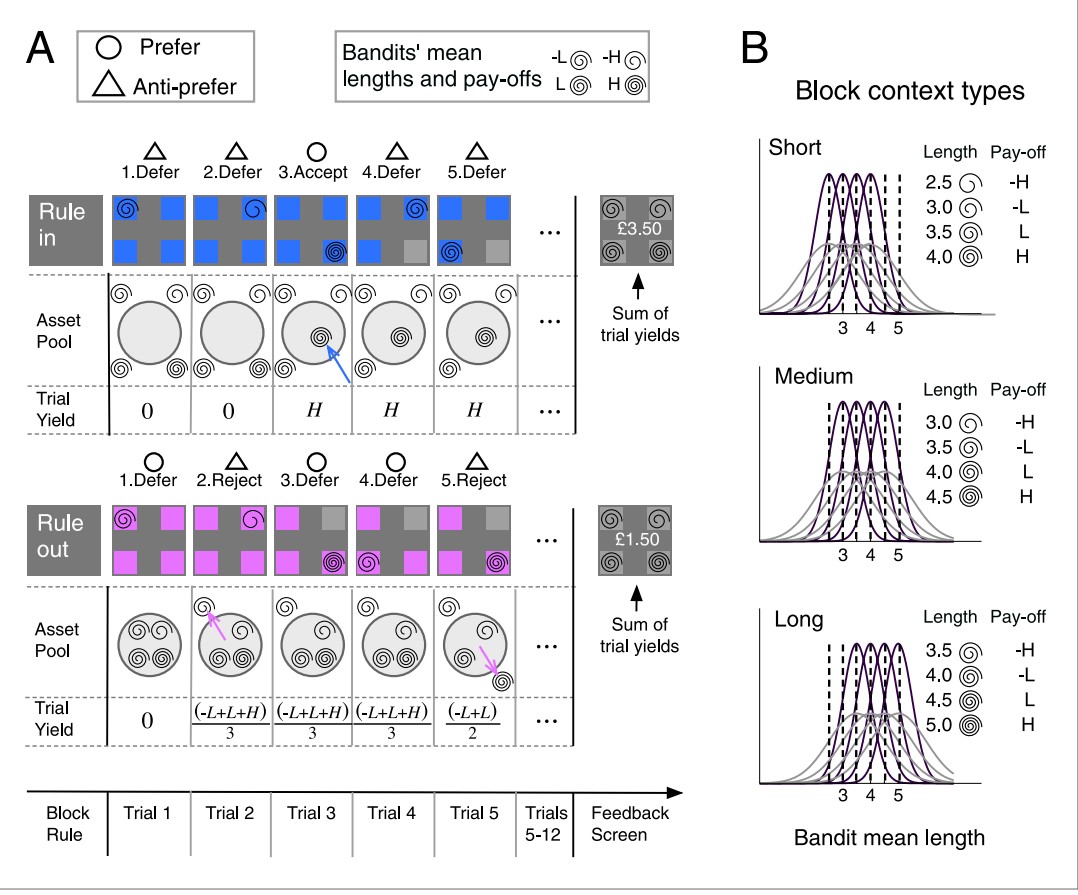

**Figure 1**. Block timeline and task design. (**A**) Upper left inset box: in the two examples, preference and anti-preference for a bandit is indicated with an open circle and triangle, respectively. Upper right inset box: showing the mapping between mean spiral length and payoff (*H* for high and *L* for low) of the four bandits in the example blocks. Upper panel: example of a rule-in block. Following an instruction screen, on each trial (grey panels) four bandits (colored boxes) were presented. A spiral in one box provided a noisy estimate of bandit mean length. Bandits that were accepted were made unavailable (greyed out) for future choices (trial 4). Accepted bandits were brought irrevocably into a virtual 'asset pool' (light gray circle) that began empty (trial 3). The per-trial yield, that is, the average of the payoffs of all bandits in the asset pool, was aggregated to provide the block-end yield. After 12 trials a feedback screen revealed each bandit's nominal length and winnings. Bottom panel: same as upper, but for a rule-out block. All bandits began in the asset pool. Rejection eliminated one bandit from the pool (trials 2 and 5). Per-trial yield reflected the average payoff of bandits not yet eliminated from the asset pool. (**B**) The bandits' length distributions could vary across 2-variance level (purple/grey). Payoff reflected the rank order of a bandit's mean spiral length within the block. The average mean length of the 4 bandits ranged from 2.5 to 5 (see 'Materials and methods') and was manipulated across 3 levels corresponding to three different context types.

associated with greater payoff). After viewing the spiral participants chose whether to *commit* to that bandit, or to *defer*. Critically, commitment engendered a sustained alteration in the per-trial economic yield, in a manner that varied according to *rule type* ('rule-in' vs 'rule-out'). On rule-in blocks, participants began the block with an empty pool of assets and the per-trial momentary yield was determined by the average payoff of all bandits committed to (*accepted*) thus far (***Figure 1A***: upper panel and 'Materials and methods'). On rule-out blocks, participants' asset pool initially included all four alternatives and the per-trial yield was the average of the payoff of all bandits not yet committed to (not yet *rejected*, ***Figure 1A***: lower panel). The total yield at the end of a block was the sum of the per-trial yields (see example in 'Materials and methods'). Per-block yield was converted to a real financial incentive via a lottery procedure at the end of each run of the experiment.

Following commitment (by acceptance or rejection) a bandit was made unavailable for future decisions and on each subsequent trial offers were drawn randomly from the bandits still in play. Thus bandits

could not be definitively accepted in rule-out blocks or definitively rejected in rule-in blocks, but commitment could be continually deferred until the end of the block. Each block used one of three *context types* that determined the average lengths of the presented spirals (short, medium, long), ensuring that participants had to learn afresh the relationship between mean spiral length and payoff in each block. This variable block-dependent mapping from spiral length to payoff (*Figure 1B*) discouraged the use of fixed strategies (such as ruling in all bandits greater than a single value across the experiment). The task is described in more detail in the 'Materials and methods' section.

## Identifying the decision variable

Our first goal was to identify the quantity (decision variable or DV) on which participants based their economic commitments. Participants could optimize performance by averaging momentary samples $s_i$ to estimate the underlying mean lengths $v_i$ and the corresponding payoffs (according to the rank-order of $v_i$), trading off speed and accuracy (or exploration/exploitation) in making economic commitments. One well-described solution to speeded choice among multiple uncertain alternatives is to respond when the accumulated evidence supporting the currently favoured alternative is sufficiently larger than that for its nearest rival (*Busemeyer and Rapoport, 1988*; *McMillen and Holmes, 2006*), that is, to compare the average spiral lengths for the current and next-best bandits. A robust approximation to this is to compare the current bandit to the mean of the other options (*Niwa and Ditterich, 2008*).

We compared the ability of these *current-minus-next* and *current-minus-average* policies (as well as of other policies, see 'Materials and methods') to predict human commitment probability across the block under different rules and contexts (*Table 1*). Although both the *current-minus-next* and *current-minus-average* provided a good fit, there was a statistical advantage for the latter (comparing negative log-likelihoods: $t_{(19)} = 3.22$, $p < 0.005$). When a decision criterion was fit to the *current-minus-average* quantity separately for rule-in and rule-out blocks (*Figure 2D*) to produce discrete model choices, they bore a striking resemblance to human behaviour, capturing the proportion, the timing, and secondary aspects of commitments for all the various combinations of rule context-type (*Figure 2A–C*).

## Brain imaging data

Next, we turned to the neuroimaging data to validate our modelling approach and measure how value was coded in medial prefrontal cortex during acceptance, rejection and deferral. We conceived of the task as a factorial design crossing rule type (rule-in, rule-out), decision (commit, defer) and the parametrically varying (signed) quantity $DV_{cur-ave}$ that indexes the estimated payoff of the available bandit under the current-minus-average policy implied by our behavioural modelling (hereafter, 'value'). We further validated this DV by testing for distinct neural correlates between the estimated average value of the offered bandit and the block reference in brain regions implicated in the maintenance of contextual information relevant to action selection, including the lateral prefrontal and parietal cortices (*Koechlin and Summerfield, 2007*) (*Figure 3A–B*).

**Table 1.** Negative log-likelihood (−*LL*; mean and standard deviation) for the eight decision variables, combing differently anchoring and integration processes

| Anchor | | Integration | | | | |
|---|---|---|---|---|---|---|
| | | No | | | Yes | |
| | $r(t)$ | $DV(t)$ | −*LL* | | $DV(t)$ | −*LL* |
| No | N/A | $s_i(t)$ | 200 ± 25 | | $\overline{v}_i(t)$ | 195 ± 26 |
| Previous | $s_i(t-1)$ | $s_i(t) - r(t)$ | 211 ± 24 | | $\overline{v}_i(t) - r(t)$ | 208 ± 23 |
| Max-next | $argmax_{j \neq i}\{\overline{v}_j(t)\}$ | $s_i(t) - r(t)$ | 183 ± 24 | | $\overline{v}_i(t) - r(t)$ | 178 ± 25 |
| Average | $\frac{1}{|S_{pres}|}\sum_{j \in S_{pres}}\overline{v}_j(j)$ | $s_i(t) - r(t)$ | 189 ± 23 | | $\mathbf{\overline{v}_i(t) - r(t)}$ | 167 ± 28 |

The best fitting DV (Anchor: *average*, Integration: *Yes*) is highlighted with bold. We refer to this DV in the text as *current-minus-average*. The second best DV (Anchor: *Max-next*, Integration: *Yes*) is mentioned in the text as *current-minus-next*.

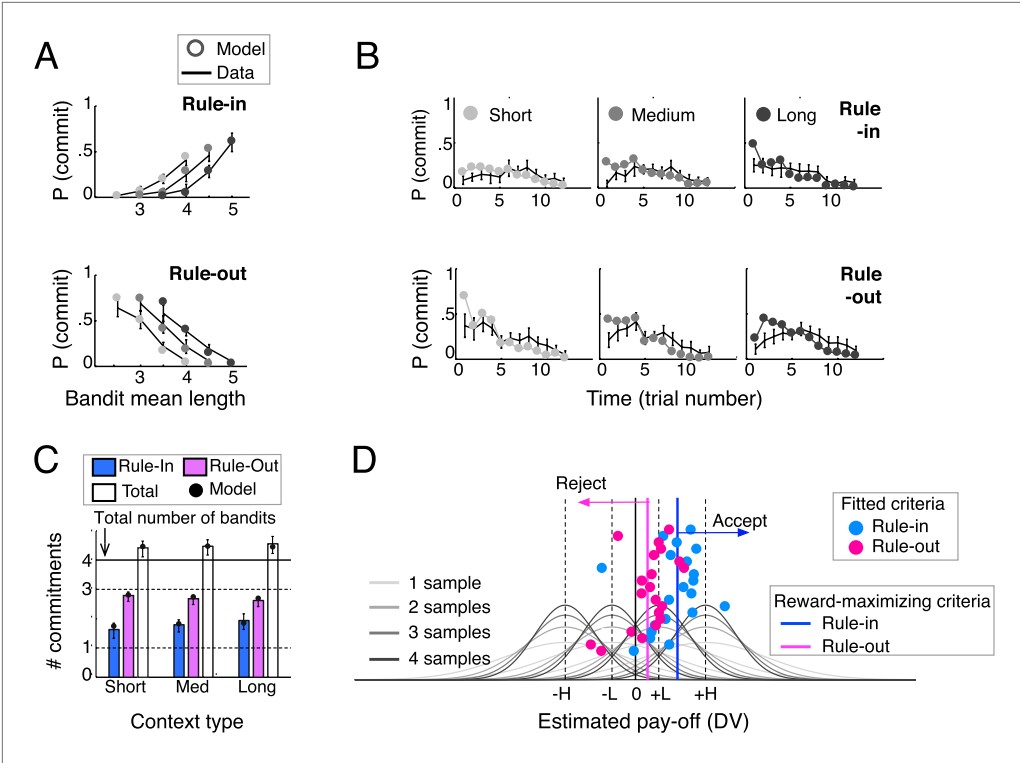

**Figure 2**. Behavioral results (N = 20) and model predictions. (**A**) Commitment probability in different contexts (short, medium and long blocks) as a function of mean bandit spiral length for rule-in (top) and rule-out blocks (bottom) and (**B**) probability of commitment as a function of trial number, context-type and rule. Black lines: human data; filled gray circles: model fits. (**C**) Mean number of commitments in rule-in and rule-out (and their sum), in the three different contexts. Moving from short to long contexts, commitments increased in rule-in ($F_{(2,38)}$ = 6.73, p < 0.01) and decreased in rule-out ($F_{(2,38)}$ = 4.95, p < 0.05). The model predicts this pattern (filled circles) by initializing the block reference to the mean spiral length in the experiment (see 'Materials and methods'), thus over(under)-estimating the DV at trial 1 in long (short) contexts. The sum of commitments exceeded the number of available bandits (4.5 ± 0.6; $t_{(19)}$ = 3.64, p < 0.005), mainly due to more than one commitments made in rule-in. (**D**) Fitted decision criteria (filled circles) did not significantly differ from reward-maximizing criteria (solid vertical lines) under the current-minus-average model for rule-in (blue) and rule-out (purple). Gray curves show the distributions of the estimated pay-off for each of the four bandits under different numbers of samples (different shades). Values larger (smaller) than the rule-in criterion provoke inclusion by commitment (exclusion by deferral). Values smaller (larger) than the rule-out criterion result in exclusion by commitment (inclusion by deferral). Bars are 95% confidence intervals (C.I.). H and L stand for bandits with high and low absolute pay-off, respectively.

## The rmPFC encodes value only when commitment is deferred

In standard paradigms involving one-shot decisions between two alternatives, BOLD signals in more rostral portions of the medial frontal cortex tend to correlate positively with the value of a preferred (or chosen) option relative to an anti-preferred or unchosen option (***Boorman et al., 2009***; ***Hunt et al., 2012***). We were thus surprised to find that no voxels in this region varied inversely with the three-way interaction between rule, decision and value (i.e., encoded decision x value inversely under the two rules). Thus, we searched across the brain for voxels that correlated with value on defer and commit trials separately (***Figure 3C***). Whereas no voxels responded to the rule x value interaction on commit trials (right panel), a prominent cluster in rmPFC was sensitive to the interaction of rule and value when participants made deferral choices (***Figure 3C***, left panel; peak at −2, 56, 18; rule x value interaction on defer trials: $t_{(19)}$ = 5.47, p < 0.00003; ***Figure 3D***). In this region, value encoding on defer trials differed from zero on rule-in ($t_{(19)}$ = 5.21, p < 0.0001) and rule-out ($t_{(19)}$ = 3.41, p < 0.002) blocks, but failed to diverge from zero when commitments were made for either rule-in (p = 0.13) or rule-out

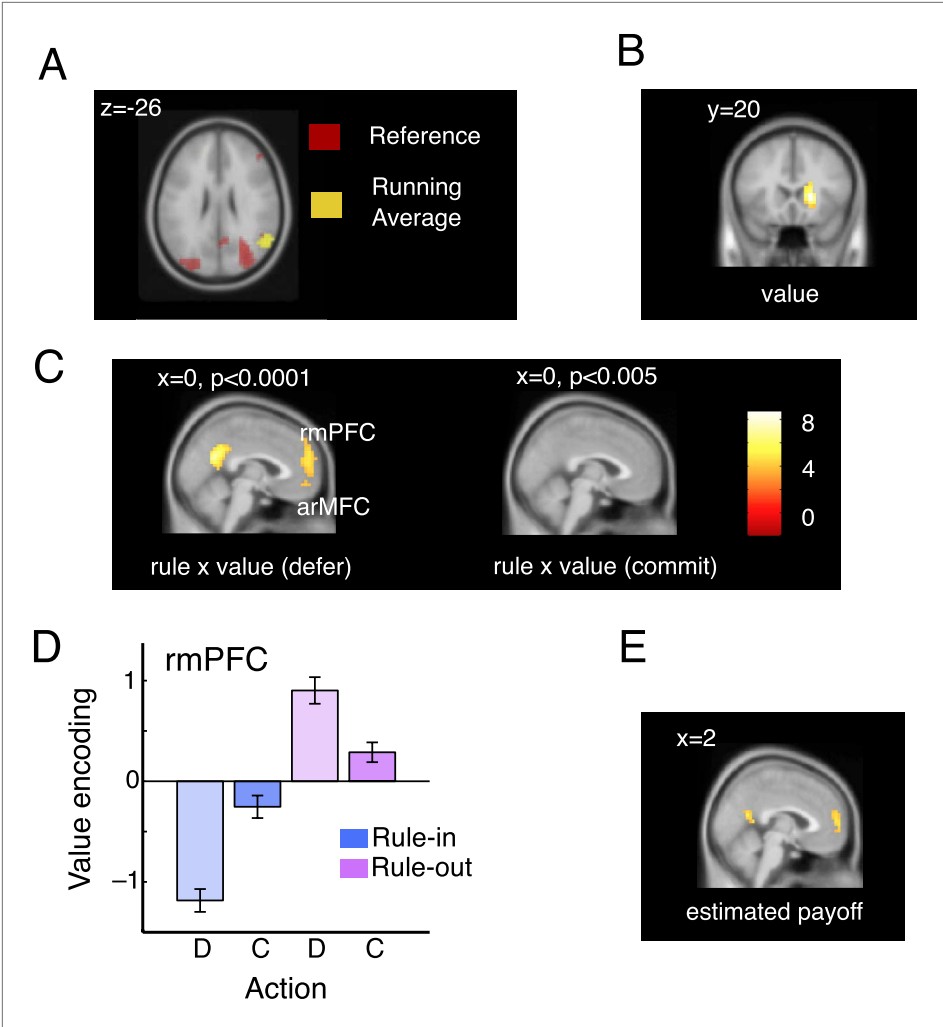

**Figure 3**. Imaging data: model validation and rmPFC. (**A**). Overlapping activations in the parietal cortex elicited by the current bandit running average (yellow; peak: 58, −60, 26; $t_{(19)}$ = 4.78, p < 0.0002) and reference (red; peak: 34, −68, 22; $t_{(19)}$ = 6.17, p < 0.00001). (**B**) In the right caudate nucleus, we also observed a representation of the difference between these two quantities, that is, voxels that co-varied with the $DV_{cur-ave}$ but did not vary according to the rule type or decision (main effect of the value signal; peak: 18, 20, 2; $t_{(19)}$ = 6.63, p < 0.00001). (**C**) Voxels responding to the interaction of rule and value on defer trials (left, at p < 0.0001) and commit trials (right, at p < 0.001). Value is encoded (in the frame of reference of the rule) only on defer trials. (**D**) Mean parameter estimates, derived by regressing bandit value on the BOLD signal from within an independently-defined ROI in the rmPFC, separately for defer and commit decision under each rule. To ensure independence, ROIs were defined individually for each participant as the peak voxel responding within the region in the remaining 19 participants. All significant voxels are visualized at p < 0.001 and survive correction for multiple comparisons across the brain. (**E**) Parameter estimates from a regressor encoding the value of the asset pool (estimated final payoff).

(p = 0.07) blocks (*Figure 3D*). A statistically indistinguishable pattern of activity was observed more ventrally in the medial orbitofrontal cortex (BA 10), a region that has been labeled arMFC or vmPFC (rule x value interaction on defer trials: peak −2, 52, −10; $t_{(19)}$ = 4.85, p < 0.00006) and where activity typically correlates the expected value during independent choices (*O'Doherty et al., 2001*; *Plassmann et al., 2007*). This sensitivity to value during deliberation but not at commitment in the rmPFC is reminiscent of single neurons in the parietal cortex that parametrically encode confidence about sensory signals but drop off precipitously at the choice point (*Roitman and Shadlen, 2002*).

## The rmPFC encodes the collective value of current assets

One possibility is that the rmPFC is involved in integrating value signals across prospects and time. If so, one might expect that it also encodes the current value of the asset pool—a quantity that signals the likely total reward that will be received at the end of the block. We calculated asset pool value in a trialwise fashion and included it as an additional predictor of the BOLD signal alongside the value, choice, rule and other nuisance factors such as the number of trials elapsed thus far in the block (see 'Materials and methods'). Asset pool value captured unique variance in the BOLD signal in an overlapping region of posterior rmPFC (peak 2, 60, 6; $t_{(19)}$ = 4.30, p < 0.0004; see *Figure 3E*). This provides corroborating evidence that the rmPFC is involved in value integration, encoding the most likely estimate of the forthcoming monetary yield to be received at the end of the block.

## The dACC responds during economic commitment

Next, we compared brain activity when decisions were made to commit or defer. Commitments in both rule-in and rule-out blocks (commit > defer) were associated with strong increases in the BOLD signal in a number of brain regions (*Figure 4—source data 1*), but most prominently in a dorsomedial prefrontal region encompassing the dACC (*Figure 4A*). Extracting data from an independently-defined ROI, we plotted parameters reflecting the average dACC response in each condition (see *Figure 4B*). Average BOLD signals differed between commit and defer decisions, with no difference according to rule type (p > 0.5) and no interaction (p > 0.9).

## The dACC encodes the value of rejection and of failure to accept

Secondly, only the dACC and interconnected bilateral anterior insular cortex (AINS) were responsive to the three-way interaction between rule type, decision and value, with a peak in activation at −6, 32, 34 ($t_{(19)}$ = 5.81, p < 0.00002; *Figure 4C*). The dACC BOLD signal correlated positively with value when participants made commitments in rule-out blocks ($t_{(19)}$ = 7.87, p < 0.000001) or deferred in rule-in blocks ($t_{(19)}$ = 4.57, p < 0.0002), but did not diverge from zero when participants committed in rule-in blocks (accept, p = 0.12) or deferred in rule-out blocks (failed to reject, p = 0.09). Note that the parameter estimates plotted in *Figure 4D* are correlations with bandit value, not raw BOLD amplitudes, and thus very unlikely to reflect cognitive demand or other nuisance factors that might conceivably vary across the block. The functional significance of this pattern of dACC activity is discussed below.

## A computational account of economic commitment

Our task emphasises two key axes that characterise economic choices. Firstly, should I definitely accept a prospect, or reject it? Secondly, should I commit now or defer my choice to later? Our behavioural findings indicate that humans made economic commitments when the *current-minus-average* DV exceeded (rule-in blocks) or failed to surpass (rule-out blocks) a relevant criterion (*Figure 2D*). Next, we used numeric simulations to specify where that criterion should be placed for rewards to be maximised, and tested how this account compared to human performance.

### Human commitment criteria maximise reward

Model-derived best-fitting commitment criteria for individual participants under the *current-minus-average* model are shown in *Figure 2D*, separately for rule-in blocks (blue dots) and rule-out blocks (pink dots). These are superimposed upon the average estimate of the value distribution for each of the four bandits (ranked low-high) as it evolved with increasing number of samples (shaded grey lines). To understand the best policy for criterion setting, we varied the decision criterion gradually as a free parameter, and plotted the reward-maximising criterion value separately for rule-in (blue line) and rule-out blocks (pink line). Critically, average human performance did not differ from that of the simulated reward-maximising account—blue and pink dots cluster around the respective lines denoting reward-maximising performance in each condition (*Figure 2D*). This was confirmed by statistical comparison of the human and reward-maximizing criteria (rule-in: $t_{(19)}$ = −1.00, p > 0.3; rule-out: $t_{(19)}$ = −0.17, p > 0.8). In other words, humans set their criterion in a fashion that maximised overall reward.

### Rewards are maximised via exclusion proneness and a deferral bias

Several aspects of criterion placement are worthy of comment. Firstly, criteria for both rule-in and rule-out blocks lie to the right of zero. In other words, participants were more prone to reject offers (in rule out blocks) or fail to accept them (in rule in blocks) than vice versa (this *exclusion proneness* corresponds to a bias to reject offers in rule out blocks, and fail to accept them in rule in blocks; we

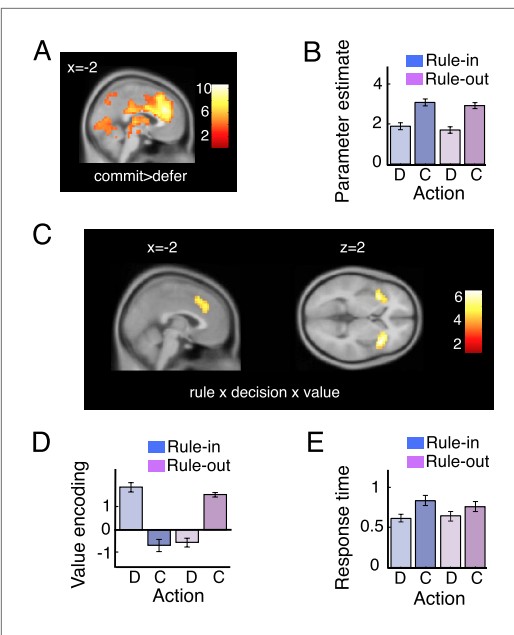

**Figure 4**. Imaging data: dACC. (**A**) Voxels responding to commit > defer, rendered onto a sagittal slice of a template brain (see also **Figure 4—source data 1**). The red-white scale shows t-values. (**B**) Average BOLD responses for defer (**D**) and commit (**C**) trials on rule-in (blue) and rule-out (magenta) blocks. (**C**) Voxels responding to the three-way interaction of rule, decision and value, in the ACC. (**D**) Bar plots showing average parameter estimates for a regression of value on BOLD activity in regions of interest (ROI) in the ACC, separately for defer and commit decision under each rule. Legend as for 3D. (**E**) Response times (seconds) were overall slower during commitment and this difference was pronounced in rule-in trials. This pattern is comparable with ACC average bold for defer and commit (**B**). Error bars are 95% confidence intervals (C.I.).

The following source data is available for figure 4:

**Source data 1**. Local maxima responding to commit > pass, at a FWE-corrected threshold of p < 0.05.

use the term 'exclusion' rather than 'rejection' to avoid confusion over terminology). This was confirmed by statistical analysis: human commitments were made more frequently (total 2.7 ± 0.3 vs 1.8 ± 0.5; $t_{(19)}$ = 6.23, p < 0.0001), and first commitment occurred earlier (trial 3.4 ± 1.3 vs 4.6 ± 1.5; $t_{(19)}$ = 3.92, p < 0.001) in rule-out blocks (**Figure 2B**). This bias helps maximise rewards because in our task participants will ideally accept (or fail to reject) only the single most valuable bandit.

Secondly, the reward-maximising criteria are not perfectly aligned for rule-in and rule-out blocks. In combination with exclusion proneness, this occurs due to an overall bias to defer commitment and thereby promote exploration. This brings the criterion closer to zero in rule-out blocks (where deferral indicates preference), and pushes it further from zero in rule-in blocks (where deferral indicates anti-preference). Indeed, participants deferred more frequently than they committed, with commitments occurring on 15% of trials on rule-in blocks and 22% of trials on rule-out blocks (with the difference reflecting the need for more commitments in rule-out). Response times were also prolonged on decisions to commit (**Figure 4E**), as if participants were overcoming a default tendency to defer ($F_{(1,19)}$ = 77.9, p < 0.0001). This effect was stronger in rule-in blocks, leading to a rule type (rule-in, rule-out) x decision (commit, defer) interaction on response times ($F_{(1,19)}$ = 27.8, p < 0.0001) with no main effect of rule type ($F_{(1,19)}$ = 1.1, p = 0.3).

## An adaptive explanation for economic 'framing' effects

Thus, in our task the reward-maximising criteria for acceptance and rejection are not perfectly aligned—the criterion for rejection is somewhat lower than that for acceptance, and this is also the case for best-fitting human criteria (*in* = 0.39 ± 0.29, *out* = 0.12 ± 0.25; $t_{(19)}$ = 3.40, p < 0.01). In other words, humans are willing to prefer an offer that they might, under a different frame, not prefer. This preference reversal in one-shot tasks would violate the rational axiom of description-invariance (**Shafir, 1993**; **Yaniv and Schul, 1997**). However, in our task this policy is the one that maximises reward, with the criteria misalignment reflecting a deferral bias that promotes exploration before commitment.

## Discussion

Previous studies have examined the neural mechanisms of economic behavior by offering participants a succession of independent gambles or unrelated consumer choices (**Daw et al., 2006**; **Plassmann et al., 2007**; **Lim et al., 2013**). Here, we devised a task in which decisions involved commitment to an asset with enduring financial consequences. Whereas previous tasks have sought to mimic the experience of a gambler choosing which one-armed bandit has the highest yield, our experiment captures that of a consumer deciding to sell an ageing car or of an animal electing whether to accept a prospective mate. This approach thus allowed us to model the factors that drive everyday economic

commitments, and to measure how brain activity differed when commitments were made relative to when they were deferred to a later time.

This approach allowed us to investigate how value was coded according to two key axes: whether decisions expressed preference or anti-preference; and whether a commitment (acceptance or rejection) was made immediately, or deferred into the future. We found a striking dissociation in the neural data; the dACC encoded value differentially as a function of the former axis (encoding value only when a prospect was anti-preferred i.e., rejected or not accepted). By contrast, the rmPFC encoded value differentially along the latter axis, covarying with value only when acceptance or rejection was deferred into the future, and not at the point of commitment.

This latter profile of activity was maximal over posterior rostromedial (rmPFC) sites in Brodmann's area nine that have been previously implicated in forecasting the value of future behaviour (*Bechara et al., 1994*), for example contributing to episodic future thinking (*Schacter and Addis, 2007*). BOLD activity in this region resembles firing rates of 'integration' neurons that build up to the point of choice and then falls away (*Roitman and Shadlen, 2002*). The wider function of the rmPFC in humans may be to integrate value across time and assets, potentially to calculate the value of prospective states or investments. Other studies have emphasised that the rmPFC may contribute to the integration of reward values across time (*Philiastides et al., 2010*; *Hunt et al., 2012*) and across goods (*Fellows, 2006*; *Lim et al., 2013*).

Modelling of behavioural performance suggested that economic commitments were made when the normalised bandit value fell above (in rule in blocks) or below (in rule out blocks) a fixed criterion. Empirically observed criteria for ruling-in and ruling-out differed, allowing the same offer to be both perceived as both 'good' and 'bad', depending on the framing of the task. Similar preference reversals due to violations of the rational principle of description-invariance have been previously described in one-shot multi-attribute choices (*Shafir, 1993*). However, in our ecologically valid task, this misalignment of the criteria for acceptance and rejection is the policy that maximises reward. Intuitively, this asymmetry reflects a default bias towards deferring commitments, and the fact that this bias shifts the criterion in opposite directions in rule-in and rule-out. To revert to a consumer example, when purchasing a house one may wish to begin with a critical eye (stringent criterion), not accepting impulsively any one property before obtaining an overview of the market and its options; but when selling a house, one might wish to be less critical (less stringent criterion) so as not to reject early offers before enough information is collected. Our finding thus provides an adaptive explanation for violations of description invariance, or preference reversal due to 'framing effects' in economic choice tasks (*Shafir, 1993*; *De Martino et al., 2006*; *Tsetsos et al., 2012*).

The dACC was sensitive to the main effect of commit vs defer, but dACC BOLD signals also correlated with value when participants rejected (or failed to accept) a prospect. In other words, the dACC is a good candidate for encoding the two biases observed in behavioural data: an overall (additive) proneness to defer, and a heightened gain of encoding value when participants dispreferred a prospect. BOLD signals may have been higher for commit than defer because participants had a higher threshold for commit decisions, leading to greater deliberation and more overall decision-related activity on these trials, consistent with prolonged reaction times observed for commit than defer decisions. We thus speculate that the two key decision biases described here on choice can be accounted for with discrete additive (deferral bias; e.g., biasing the deferral threshold) and multiplicative (exclusion proneness; e.g., modulating the rate of accumulation for negative values) parameters under mechanistic models of choice (*Bogacz et al., 2006*; *Krajbich and Rangel, 2011*), and that the dACC may implement these biases in our task. These findings concur with the previously-noted sensitivity of dACC to value difference, but also with reports that the dACC and co-activated anterior insula are prominent among those regions that signal a switch away from a current or 'default' task (*Hyafil et al., 2009*) or status quo position (*Fleming et al., 2010*).

Of note, the two biases reported here bear a striking resemblance to the previously described tendency to favour a currently-endowed or status quo economic position (deferral bias) and loss aversion, the tendency to weight potential losses more heavily than potential gains in economic choice (exclusion proneness). In other words, one possibility is that three key economic suboptimalities catalogued in one-shot decision tasks—framing effects, endowment effects, and loss aversion—are all 'rational' biases when a more ecologically valid task is employed, in which decisions do not all have independent consequences (*Erev and Roth, 2014*; *Fawcett et al., 2014*).

In summary, asking participants to commit to economic alternatives revealed a striking dissociation between the dACC and rmPFC, two brain structures whose contribution to economic choice in the

primate remains highly contentious (*Kable and Glimcher, 2009*; *Rangel and Hare, 2010*; *Rushworth et al., 2011*). BOLD signals in the rmPFC and dACC both correlate with the relative value of two prospects under consideration, but do so with opposite sign, which has led researchers to puzzle over their unique contributions to decision-making. Some theories have proposed that rmPFC and dACC encode the value of stimuli and actions respectively (*Rudebeck et al., 2008*), or the value and saliency of stimuli (*Litt et al., 2011*), or the value of choosing a food item relative to that of foraging elsewhere (*Kolling et al., 2012*). However, past work has precluded the segregation of neural activity on trials where a commitment was made, relative to those where commitment was deferred into the future.

Our findings suggests that the rmPFC is involved in integrating value in the service of prospective states, whereas the value coding in the dACC is relevant to whether a currently-available prospect is preferred or not. As such, these findings support the view that the hierarchy of control signals observed in the lateral prefrontal cortex is similarly instated in the medial prefrontal cortex (*Kouneiher et al., 2009*; *Summerfield and Koechlin, 2009*).

## Materials and methods

### Participants

21 healthy right-handed adults (mean age = 26.8 ± 3.7 years; 10 females) gave informed consent to participate in two experimental sessions (a practice session, and an fMRI session) conducted on different days, and were compensated £40 plus up to £20 in performance-dependent bonuses. One participant was excluded from subsequent analyses because of failure to comply with the task instructions.

### Stimulus and task design

On practice and fMRI sessions, participants performed a task on which they decided whether to accept or reject stimuli ('bandits') on the basis of visual signals (spirals). On each trial ($t$), they viewed a spiral of variable length ($s_i(t)$) that appeared in one ($i$) of four pink or blue boxes ('bandits'), placed in the four quadrants of the screen. Bandits were randomly assigned the following four *payoffs* ($\mu_i$):£15/24, £5/24, −£5/25, −£15/24 corresponding to the symbols *H(high), L(low), −L, −H* in the figures. Although payoffs were fixed, the mapping from payoff to the nominal mean spiral length changed block-by-block, so that bandit payoff was relative to the other spiral lengths observed in any given block. Each bandit's spiral length was drawn from a normal distribution $N(v_i, \sigma_i)$ where $v_i$ was related ordinally to the payoff $\mu_i$ of the $i$th bandit, and $\sigma_i$ stood for the standard deviation of the distribution. There were six possible mean spiral lengths,$v_i$, in the experiment: 2.5, 3.0, 3.5, 4.0, 4.5 and 5 (*Figure 1B*). Splitting these six means into contiguous groups of four resulted in three different *context types*, presented in pseudorandom order: (a) 'short' blocks comprised of bandits with mean lengths (from least to most valuable) of 2.5, 3.0, 3.5 and 4.0, (b) 'medium' blocks with bandits with spiral length means 3.0, 3.5, 4.0 and 4.5, and (c) 'long' blocks with bandits with spiral length means 3.5, 4.0, 4.5, and 5.0. Additionally, on each trial, two of the bandits had a standard deviation of 0.5 while the other two had a standard deviation of 1.0 (*Figure 1B*). Since the only determinant of a bandit's payoff, $\mu_i$, was the rank of its mean length within the block context, the bandit whose spirals had mean length $v_i = 4.0$ would be second in rank and be worth $\mu_i = $ £5/24 (*L*) in a 'short' block and would rank third, yielding $\mu_i = −$£5/24 (*−L*) in a 'long' block. This approach precluded fixed strategies such as accepting or rejecting all bandits below or above a single spiral length. The presentation order of the blocks was pseudo-randomized such that there where 4 blocks of each length type within a scanner run (12 blocks per run).

Trials occurred in 48 blocks distributed in four runs. The reward assignment of the task varied according to the rule type. On 'rule-in' blocks, a spiral was presented in one of the four bandits and participants chose whether to 'accept' that bandit, or 'defer'. The per-trial yield was equal to the average payoff of all the bandits ruled in up to that point: $Y_{in} = \dfrac{\sum_{i \in IN}(\mu_i)}{|IN|}$ at trial $t$, with $IN$ standing for the set of the bandits that had been ruled-in up to $t$ We refer to the items thus far ruled-in as the *asset pool*. On 'rule-out' blocks, participants chose whether to 'reject' that bandit or to 'defer', and their per-trial yield was equal to the average payoff of all bandits not yet ruled out at that point $Y_{out} = \dfrac{\sum_{i \notin OUT}(\mu_i)}{4-|OUT|}$, with $OUT$ being the set of bandits that had been ruled out up to that point. Items not yet ruled-out are contained in the virtual asset pool. For both rule-in and rule-out the block-end

yield was the sum of the per-trial yields of all trials (with the last trial of the block indicated by $k \in [4, 12]$), rounded to the nearest half or whole number: $\sum_{t=1}^{k} Y_{rule}(t)$. For example, in a rule-in block, if after $t = 8$ trials the participant had ruled in bandits with payoffs £5/24 and £−5/24, the yield on that trial would be: $Y_{in}(8) = £\left(\frac{15-5}{24}\right)/2$. Similarly, in a rule-out block if at trial t = 8 participants had rule out bandits worth £−15/24 and £−5/24 the yield on that trial would be the average of the remaining bandits: $Y_{out}(8) = £\left(\frac{15+5}{24}\right)/2$.

Payoffs were calibrated so that selecting no bandits, or selecting all the bandits, would bring per-trial payoff to zero.

## Time course of a block

Rule-in and rule-out blocks were interleaved in pseudorandom order. On each experimental run (corresponding to 12 blocks) 6 blocks of each rule type were presented. Each block began with a fixation cross presented for 2 s under the words 'Rule In' or 'Rule Out' that announced the rule type of the block. Immediately after, four coloured boxes, one for each bandit, were presented in the four quadrants of the screen against dark grey background. The initial color of all bandits was either blue or pink, indicating the rule type of the block (counterbalanced across participants). Bandits that had not yet ruled in or out, and were thus available for future decisions, were set *active* and kept their initial color. On each trial, a spiral of variable length appeared randomly at the centre of one of the active boxes (bandits). Participants had a maximum of 2 s to either commit (rule in/rule out) to the bandit under offer or to defer. Committing to a bandit altered its status to inactive and resulted in the corresponding box being colored gray until the end of the block. In both rule-in and rule-out blocks, participants indicated their choice with a key press (practice task) or button press (fMRI session). Spirals disappeared from the screen immediately after participants pressed a button. Failure to respond within the deadline (2 s) resulted in forced commit or defer choices made automatically and randomly. After the response was registered, the next spiral was presented after a delay between 1 and 3 s. A block ended either when there were no more active bandits left or after 12 trials.

## Reward screen

At the end of the block the four bandits (turned into or) remained gray between 3 and 5 s (jittered). Immediately after, participants received a feedback screen indicating their earnings. Inside the box, corresponding to each bandit, the mean spiral length was shown together with the time that had elapsed (presented as a filled pie chart) before participants had made a committing decision. Centrally on the screen, the block-end monetary yield was shown (*Figure 1A*). In the scanning session the feedback screen remained present until 85 s from the beginning of the block (presentation of the fixation cross) had elapsed. Thus, in the worse case scenario where all presentation and response events took the maximum possible time, the feedback screen stayed on for 15 s. On the other hand, in the behavioral sessions participants could advance by hitting any button. After an inter-block interval of between 2 and 4 s (jittered) the new block began. Every 12 blocks (i.e., at the end of one scanner run) participants viewed a 'wheel of fortune' consisting of 12 segments (1 for the block-end reward of each block) coloured proportionally to each block-end yield (red to green for negative to positive respectively). Participants pressed a button to spin the wheel of fortune and then another button to stop it. On average, participants won £3.0 per block (SD = 0.7). There was no significant difference between the rewards in the rule-in and rule-out trials ($t_{(19)} = 0.50$, p = 0.620). The value of the obtained segment was shown on the screen and then added to participants' earnings obtained in previous runs. The overall earnings in a session could not exceed £10.

## Decision Variables

For parsimony, we considered decision variables (corresponding to the payoff estimate of each bandit) that avoid latent or hierarchical inferences that might take place in the course of the experiment. These variables varied across two features: (a) *integration* or computation of the *absolute value* of a bandit through averaging the spiral lengths presented thus far for that bandit and (b) *anchoring* of the absolute value (integrated or not) for each bandit to a reference value that represents the value of the

other bandits in the block. For any given variable, integration (averaging) was either present or absent. Absence of integration means considering the momentary length presented at a bandit while ignoring all previous spirals encountered before at the same bandit. Anchoring could be omitted ('absolute' DV) or manifest itself by means of subtracting an implicit reference value from the absolute value of a given bandit. The reference value ($r$) changes trial-by-trial as new information is presented. Thus, at trial $t$, $r(t)$ could be the absolute value (integrated or not) of the temporally previous bandit, the absolute value of the next-best bandit, or the average of the absolute values of all bandits in the block. For DV's that involved anchoring the reference on the very first trial of a block was set to $v_{prior} = 3.75$, which reflected the mean spiral length in the experiment given the distributions in *Figure 1B*. The formulas associated with each DV are given at *Table 1*. We define with $S_i^t$ the set of trials, up to trial $t$, in which spirals were presented at bandit $i$. The average value estimate for bandit $i$ at trial $t$ is

$$\overline{v}_i(t) = \frac{1}{|S_i^t|} \sum_{j \in S_i^t} s_i(j),$$ with $s_i(j)$ being the spiral length in trials ($j$) where value information was presented

for bandit $i$. In *Table 1*, we refer with $S_{pres}$ to the set of bandits for which value information was presented at least once in the block.

## Model comparison and fitting

Using logistic regressions, we assessed how well each of the eight DV's of *Table 1* predicted participant's probabilities to commit. Separate regressions were performed for rule-in and rule-out trials:

$$P_{in}(commit) = \Phi(b_{in} + c_{in}DV)$$

$$P_{out}(commit) = \Phi(b_{out} + c_{out}DV)$$

where $\Phi$ is the cumulative normal function. These logistic fits assume that commitments are made once the DV exceeds a criterion threshold (reflected at the intercept, $b$). The negative log-likelihoods (summed for rule-in and rule-out) for each DV were calculated for each participant and compared using $t$ tests. In order to generate discrete choices for the exact same sequences that participants saw in the experiment, the best-fitting $DV_{cur-ave}$ was used to fit the probabilities of commiting in 24 conditions: four bandits' rank-ordered lengths (–H, –L, L, H) × 3 context types (short, medium, long) × 2 rule types (rule-in, rule-out). The purpose of generating discrete choices using the $DV_{cur-ave}$ was to assess the adequacy of the model in predicting qualitative aspects of the data (*Figure 2A–C*). The only free parameters were the decision criterion in rule-in and rule-out. We adopted a Maximum likelihood parameter estimation approach. This simple model was also compared to two models with additional free parameters: (a) a leaky averaging model that implemented an exponential moving average when calculating the absolute integrated value of each bandit, (b) a leaky averaging model, in which the initial value of the reference ($v_{prior}$) was a free parameter. Bayesian information criterion analyses revealed no advantage of the two extended models over the simple $DV_{cur-ave}$ one.

## Behaviour

Each participant completed a one-hour long practice session outside the scanner between 2 weeks and 3 days prior to his or her scanning session. The experimental task completed during practice was identical to the task performed in the scanner, with two exceptions. First, participants in the practice session made committing choices by pressing the 'm' button on a standard PC keyboard and made defer choices by pressing the 'k' button, as opposed to using the a MRI-safe response pad in the scanning session (left or right button, counterbalanced across participants). Second, participants during practice controlled the time at which they advanced through the feedback screen (and on to the next block) by pressing any computer keyboard button, instead of having to wait a specified amount of time as was required during the fMRI session. Stimulus presentation was conducted and behavioral responses were acquired using Matlab 7.4 (MathWorks, Natick, MA) and Psychophysics toolbox extension (*Brainard, 1997*; *Pelli, 1997*) on a standard PC.

## Behavioural analysis

For behavioral analysis an alpha value of 0.05 was used except where otherwise noted and all tests were two-sided. Dependent variables in different analyses involved the probability of committing

(defined as the number of commitments divided by the number of trials in which a certain bandit was under offer), the trial number of the first commitment, and the per-block total number of committing decisions, for different bandit means and across different rules (rule in/rule out) and context types (short/medium/long).

## fMRI data acquisition

Neuroimaging data was acquired with an echo planar imaging (EPI) sequence on a Siemens Tim Trio 3.0T scanner with a 32-channel head coil. Scans were acquired with a repetition time (TR) of 2000 milliseconds (ms), an echo time (TE) of 30 ms, a voxel size of 3 × 3 × 3.5 mm, and a flip angle of 90°. One image volume consisted of 36 adjacent slices of 3 mm thickness, allowing most of the brain (not all of the cerebellum) to be fit into the field of view. Each of the four experimental runs in the fMRI session lasted approximately 17 min and participants were given 1 min of rest between blocks. Additionally, a structural scan using a Magnetization Prepared Rapid Acquisition Gradient Echo (MP-RAGE) sequence (RT of 2040 ms, TE of 4.7 ms, flip angle of 8°, voxel size of 1 × 1 × 1 mm) was acquired immediately following the four experimental blocks. This structural scan took approximately 6 min and resulted in a total scanning time of approximately 78 min.

## fMRI preprocessing

fMRI data was preprocessed according to a standard pipeline in SPM8 (Statistical Parametric Mapping; www.fil.ion.ucl.ac.uk/spm). Preprocessing of the imaging data included correction for head motion and slice acquisition timing, followed by spatial normalization to the standard template brain of the Montreal Neurological Institute (MNI brain). Images were resampled to 4 mm cubic voxels and spatially smoothed with a 8 mm full width at half-maximum isotropic Gaussian kernel. A 128 s temporal high-pass filter was applied in order to exclude low-frequency artifacts. Temporal correlations were estimated using restricted maximum likelihood estimates of variance components using a first-order autoregressive model. The resulting nonsphericity was used to form maximum likelihood estimates of the activations. Importantly, SPM8 orthogonalises sequentially-entered regressors in the design matrix, but we ensured that this option was turned off for all analyses. All statistical analysis of imaging data included, in addition to regressors of interest, nuisance parameters encoding (i) the fixation screen that signaled block onset, (ii) the reward screen that indicated monetary earnings, (iii) an instruction screen stating rule type, and (iv) a regressor encoding the 'grey' period at the end of any blocks in which all bandits had been ruled in or out or the maximum number of trials had been reached, as well as six regressors coding movement parameters estimated from the realignment stage of preprocessing.

For comparing parameter estimates across the experimental conditions, independent samples $t$ tests were used, while one-sample $t$ tests were employed to assess whether these parameter estimates differed significantly from zero. All statistical anlayses reported in the text were corrected for multiple comparisons across the entire brain using a clusterwise threshold of $p < 0.05$, although plots are rendered at $p < 0.001$ uncorrected. To ensure independence, bar graphs were generated using a 'leave one out' procedure, as follows: (1) the ROI from the group analysis at a threshold of $p < 0.001$ was used to define a search area, (2) each subject was set aside in turn, and the peak voxel sensitive to the statistical comparison for the remaining 19 subjects was identified; (3) the (independent) contrast value for subject s was logged; (4) bar graphs and accompanying stats were produced using only independent contrast values.

## fMRI analysis

We conceived of our experiments as a factorial design crossing factors (i) rule (rule-in vs rule-out), (ii) decision (commit vs defer) and (iii) value ($DV_{cur - ave}$) as a parametric regressor encoding the value estimate of a bandit under offer. These regressors, plus their interactions (seven regressors total) were entered into the design matrix alongside additional regressors encoding (i) the reference value for that block ($r$ defined in *Decision Variables*), (ii) the trial number (1–12), that is, the time elapsed since the start of the block, and (iii) estimated aggregated yield, that is, the estimate of the most likely cumulative payoff for that trial, given the history of bandits and decisions. This was calculated by accumulating estimated per trial yield of the bandits included in the assets pool (for each bandit defined as the difference between the its average length estimate, $\overline{v_i}$, and reference value, $r$). In addition to the regressors of interest we included a series of nuisance regressors (see above). In a subsequent analysis (*Figure 4B,D*), we additionally extracted the BOLD timeseries from the peak group response to rule x decision x value in the dACC (−6, 32, 34) and included it (and its interaction with defer vs commit) as an additional regressor.

## Acknowledgements

We thank Tim Behrens, Demis Hassabis and Etienne Koechlin for comments on a draft manuscript. This work was funded by a European Research Council award to CS.

## Additional information

### Funding

| Funder | Grant reference number | Author |
| --- | --- | --- |
| European Research Council | ERC 281628 - URGENCY | Christopher Summerfield |
| National Institutes of Health | NIH R01MH097965 Subaward 13-NIH-1039 | Christopher Summerfield |

The funders had no role in study design, data collection and interpretation, or the decision to submit the work for publication.

### Author contributions

KT, Conception and design, Acquisition of data, Analysis and interpretation of data, Drafting or revising the article; VW, Conception and design, Contributed unpublished essential data or reagents; SPS, Conception and design, Acquisition of data, Analysis and interpretation of data; CS, Conception and design, Analysis and interpretation of data, Drafting or revising the article

### Ethics

Human subjects: All participants gave informed consent to participate in the experiment, agreeing also that we would store anonymously their data, analyse them, and publish the corresponding results in peer-reviewed journals. Ethical approval was provided by the local committee in Oxford: NRES Committee South Central–Oxford A, identifier 09/H0604/11. All procedures accorded with the Declaration of Helsinki.

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
