## [Decision Letter]

Thank you for sending your work entitled “Neural mechanisms of economic commitment in the human medial prefrontal cortex” for consideration at *eLife*. Your article has been favorably evaluated by Eve Marder (Senior editor) and 3 peer reviewers.

The Senior editor and the reviewers discussed their comments before we reached this decision, and the Senior editor has assembled the following comments to help you prepare a revised submission.

While all of the reviewers found significant value in the experimental part of your work, the consultation among the reviewers revealed an essential difference of opinion about the value of the model and its analysis. A number of proposals surfaced in the consultation session, but these are summarized briefly as two options that I can offer you: Option 1) Strip down the paper to its essential and interesting experimental component, resulting in a strong and far simpler paper. Then, at your leisure, develop a second paper for a computational journal fleshing out how different competing models satisfactorily or not account for the data.

Option 2) More fully develop the modeling component so that you do a fair comparison of how a number of different models capture the essential features of the data.

The most extreme statement of this position comes from this comment from the consultation session, “If they do want to decide how best to model their empirical data, my feeling is that this would make a decent second paper for a modeling journal. To do that right they ought to consider, at the least, the main competitors to the DDM: The Cisek Model and the Stuphorn Model, also they might consider Usher-Mclelland. And of course if they want to do a really good job they should include a normative benchmark like the Beck/Ma/Pouget Bayesian modelling process. Were they to fit all of these that would be an incredibly interesting exercise. My guess is that (looking at their data) Pouget will way outperform the others.”

As it stands now, there is consensus that the experimental work is more satisfying than the modeling work, and that the latter either needs to be removed or strengthened.

Ordinarily we don't include the full reviews in this consensus review, but in this case, I think it might be instructive for you to see where the reviewers were coming from initially, as long as you understand that their initial positions and attitudes all changed during the consultative discussion. The Minor Comments are those that you would ordinarily see verbatim in an *eLife* review. Some of these comments will obviously be irrelevant if you take the simpler Option 1.

Reviewer #1:

The paper by Tsetos and colleagues represents an interesting and important contribution to decision neuroscience. As the authors point out, the psychological and neural mechanisms underlying decisions involving long-term commitments and interrelated choices are largely unknown and understudied. The authors have designed a novel and suitable decision paradigm in which to examine this type of choice. Moreover, their use of computational modeling methods at the behavioral level and the integration of those computationally defined parameters and values into the neuroimaging analysis yields results at both the descriptive and mechanistic levels. In general, the work is great and I think it will be quite influential in the field. There are a few puzzling aspects of the results, especially with regards to the specificity of value (DVcur-DVave) representations in the brain for different contexts and choice types. Puzzling results are not at all bad, quite the opposite as they often promote better understanding. However, there is one straightforward analysis that the authors have not presented in the paper, but could readily apply to the current data, and may be relevant to this issue. I outline this analysis and present two minor points below.

The value term used in the neuroimaging analysis is always the difference between current and average payoff estimates. This is a perfectly reasonable definition of value and numerous fMRI papers have reported that such a difference term is reflected in mPFC, PCC, or striatal BOLD signals. However, there are also a number of fMRI (e.g. [38]; Wunderlich et al., 2009) and single unit recording papers (e.g. [35]) that describe BOLD activity or firing rates as correlating with offer or chosen values rather than or in addition to difference values. Therefore, I wonder whether there might be regions in the brain that represent DVcur as opposed to DVcur-DVave and whether those regions also show choice and context specificity? Although testing this idea would require a small amount of additional work, I believe either positive findings or null results will provide a basis for a more complete interpretation of the currently reported results.

Minor Comments: In the Introduction, [38] is described as a binary food choice task. It is not, the participants bid money in an auction to receive the foods. The [30] paper cited later in the Introduction is a good example of a binary food choice task the authors could cite here instead.

There is some disagreement in parameter values between the main text and SI. In the main text, the y-out value is listed as 7.19+/-0.35, but in the SI y-out is mislabeled as y-in and the SD is given as 2.35. These inconsistencies/typos should be corrected.

Reviewer #2:

I really enjoyed reading the paper “Neural mechanisms of economic commitment in the human medial prefrontal cortex” by Tsetsos and colleagues. The work is elegant and uses a sophisticated approach that combines modelling with neuroimaging data to address an important issue in decision theory (i.e. commitment vs. deferral). I am already familiar with this paper since I have reviewed it for a previous submission to a different journal. Since their original submission the authors have done a great job in reanalysing their data so as to address the concerns I had at the time. Furthermore in this current submission the authors have also extended their work to provide an intriguing explanation of how the framing bias observed in many classical decision tasks might be adaptive (increase reward maximisation) in the context of a more ecological task in which decisions are not taken in isolation. In this regard, I suggest that the authors look at the recent paper by Erev and Roth (just out in PNAS) that puts forward a similar case for reward maximisation in the context of learning. In this version of the manuscript Tsetsos and colleagues have also fitted a drift diffusion model to their data so as to provide an algorithmic description of the computational process preformed in PFC. I am happy to recommend this paper for publication in *eLife* and I am sure it will spark great interest in the community.

Minor Comments:

In its current form the paper is really dense in terms of results and therefore a few more lines in the Discussion section which recapitulate the main points and their significance would be of great help to many readers.

Reviewer #3:

Honestly, I struggled with this paper at a couple of levels and wanted to share with the authors what are somewhat conflicting views of the paper as written. At its core, the paper seeks to study a kind of decision that has not been too heavily studied within the area of decision neuroscience; the study of long-term commitments that yield variable income streams. Of course a number of papers have been published that look at 'investing' in stock-like objects (Cami Kuhnen was probably the first to do this a decade ago) that yield income on each trial of an experiment, but in most of those experiments subjects can change their 'investments' on each trial. A number of people have also looked at foraging decisions, which also have a similar flavor. In those experiments (and Ben Hayden's postdoc papers are probably the best early example) subjects have to decide whether to stay and harvest declining rewards or to leave and search for new, possibly better, rewards. In these experiments subjects do need to make a commitment, because deciding to switch costs them something in time and reward rates. So in these studies (which include work by Newsome's group and a lovely recent paper from Rushworth's group) there is a commitment being studied, but it is a costly rather than an irreversible commitment.

So I would have to say that the key novelty of this paper is not its study of commitment per se, but rather what is fairly unique is that it is a study of commitments that are 'irreversible' (at least for the remaining trials of the round).

Still, that is a novel area and the authors should get clear credit for exploring this new area. One could complain that they have overstated the novelty of the study by not acknowledging that studies which impose costs for 'changing your mind' (as in the matching law experiments of Newsome and Co, the foraging experiments of Hayden and Platt, or of Rushworth and Co.) are studies of commitment, but those are criticisms of exposition rather than core content.

With regard to core content, the most interesting result in the paper is the finding that 'committing' vs 'passing' yields differential activation in the ACC. That is an interesting finding and one that accords well (as noted in the paper) with previous findings by Rushworth's and Platt's groups. But even with regard to this finding, I was torn. The task that the authors used is, at heart, quite complicated. The subjects are asked to address a set of 4 uncertain bandits and to decide whether and when to commit to those bandits. So a number of phenomena are happening in parallel. First, the subjects face uncertainty which diminishes with each sample. Second, the subjects face bandits of different values. Finally, the subjects commit and face two commitment rules. So in essence one has a pile of variables moving in the task: value, certainty + probability of reward, rule, and a diminishing horizon to the end of the block. The subject adds to these her two decisions. And I think that there is enough going on these that one can’t be completely certain what is being represented at any point in time. That doesn't by any means make their findings uninteresting, not in the least, but it does make the experiment hard to interpret. Does the fact that the anterior MPFC responds more strongly in pass trials than on commit trials tell us about conditional value coding as the authors suggest or about something related to the decision other than value coding? It’s just hard to be certain.

The authors try to help us resolve this ambiguity by fitting their data with two decision rules and selecting the better fit of the two (current-minus-average) as a benchmark of sorts. But one cannot help but worry that the model they use is somewhat arbitrary. And here the fits were not has helpful to me as I might have hoped. The fits certainly capture the trends in the data, but in a subjective way. I really didn't know if the fits were 'good' in some objective sense.

Finally, the paper settles into a drift diffusion model of what the authors suggest might be the underlying process. Drift diffusion models are often used in this way, fit to the data as a demonstration that they can capture variance in a decision paper, but I wasn't honestly sure what the DDM added here. The DDM doesn't enhance our understanding of the scanner data, I don't think. And it isn't compared with any other reaction time model so the paper isn't arguing that the DDM is superior to some of other set of models for reaction time commitment decisions.

Specific Suggestions: Going forward, the paper would be improved if it began with a very clear statement of what is novel in this work. That begins, of necessity, with a clear statement of what a 'commitment' is. What is a commitment? To an economist it is a cost of some kind imposed for future 'switches'. That cost could be a few seconds as in the old Baum-Herrnstein matching law experiments, something more complicated as in the Hayden Platt experiments or a financial cost as in some of the studies of people trading in stock markets. This is a very high cost commitment and that is novel.

Next, we really need a normative theory of what a commitment does to the value of an option. That's not too hard. The model needs to ask, on each trial, what is the value of committing (on average) under the current conditions. What is the value of passing. These are aggregate discounted expectations, so there is some subtlety here, and risk aversion must be taken into account, but this is pretty standard stuff from finance 101. Then one needs to try and relate these standard finance assessments to neural activation. What one needs to show is that these standard normative models FAIL to account for what is going on and that some alternative framework exceeds the predictive neural power of this standard benchmark model.

Minor Comments: The authors use the word normative in an odd way and this leads them, occasionally, to say things that seem false. The note, for instance, Shafir's axiomatic proof about preference reversals and then go on to argue that their data show that under these conditions such preference reversals are normative. This is super tangled. The normative question is: are there times when you would rule out/rule in asymmetrically in this task. That really calls for an axiomatic proof, which would be quite do-able. But instead they use a simulation to try and show that they get good maximization with an asymmetric rule. This gets confusing fast. I would urge the authors to try and do the axiomatic proof for their environment or to just stay away from axiomatics altogether. Mixing them haphazardly with numerics is just making the story hard to follow.

---

## [Author Response]

In the revised submission, we followed your suggestion (1), i.e. to strip the paper to its empirical findings, removing the diffusion modeling component. Although our original aim was to use the diffusion model as a descriptive tool to jointly capture patterns in mean choice, response times and neural activity (rather than to claim the superiority of the diffusion model over more realistic counterparts), we do agree that mechanistic modeling and model comparison issues can and should be addressed in a separate paper. Therefore, we have removed all references to the diffusion modeling part from the manuscript and eliminated the relevant figures. Whilst there a few references to a “model” do remain, these pertain to the current-minus-average model of value coding, which was unrelated to the sequential sampling assumptions that the reviewers called into question.

Removing the sequential sampling modeling somewhat weakens the claim that the two behavioural biases (deferral bias and exclusion proneness) are dissociable at the computational level. Accordingly, we have toned down our language in asserting this claim; nevertheless, in the Discussion section we continue to call upon the neural data from the dACC (Figure 4) which show both a main effect (of commit vs. defer) and an interaction exclude/include x value to support the argument that there are separable additive and multiplicative biases, and that the dACC signals encode these biases.

Finally, as per editorial request, we removed the Supplementary Information and incorporated condensed parts of it to the main text (mainly as Materials and methods while some of the secondary results are now concisely given in figure captions).

To facilitate your processing of our revision, we outline below the main structural changes in the paper. We also point to minor changes we did to correct lapses and omissions, based on some of the reviewers’ constructive suggestions.

The whole section “Mechanistic model of commitment and deferral” and the relevant subsections were removed.

The Discussion part was modified to reflect the removal of the diffusion modeling. One full paragraph was removed and we instead added a new paragraph. This new paragraph, based on the qualitative patterns in dACC, mentions the possibility that the biases in our task are computationally dissociable.

Both reviewers 1 and 3 asked for clarification about the term “commitment” and noted that part of the interest of our task is that decisions cannot be reversed.

We have thus added to our definition of this term in the Introduction accordingly, and clarified that in our task a decision to accept or reject is final. We have taken reviewer’s 1 suggestion and replaced in the Introduction the reference to [38] with [30].

Regarding reviewer’s 2 suggestion about complementary fMRI analyses we would like to point to the new Figure 3 (A and B) and the corresponding caption. There, we show the distinct neural encoding of the average value of the bandit under offer as well as the encoding of the block’s reference value.

Following reviewer’s 2 recommendation, we now cite [11].

We have replaced the term “normative” with the term “adaptive” or “reward-maximizing” in several parts, taking on board reviewer’s 3 concern that the term normative might lead to misunderstandings.

We have removed Figures 5 and 6 that corresponded to the diffusion modeling part. Note that Figure 5D, showing the average response times in the task, has been moved to Figure 3.

We completely removed the Supplementary Information.

Figure S1B is now Figure 2. Ex-Figure 2 is Figure 2. We removed Figure S1A because it was redundant given Table S1 (currently Table 1). We removed Figure S1C because it conveyed information of secondary importance.

Figure S2 has now moved to Figure 3. Also Table S2 is now submitted as source data linked to Figure 4.